# Molecular and epistatic interactions between pioneer transcription factors shape nucleosome dynamics and cell differentiation

Rémi-Xavier Coux [1,2], Agnès Dubois [1], Almira Chervova[1], Inma Gonzalez [1], Sandrine Vandormael-Pournin[1,2], Michel Cohen-Tannoudji [1,2] ✉ & Pablo Navarro [1] ✉

Pioneer transcription factors (TF) bind nucleosome-embedded DNA motifs to activate new regulatory elements and promote differentiation. However, the complexity, binding dependencies and temporal effects of their action remain unclear. Here, we dissect how ectopic induction of the pioneer TF GATA6 triggers Primitive Endoderm (PrE) differentiation from pluripotent cells. We show that transient GATA6 binding exploits accessible regions to decommission enhancers and promote pluripotency gene silencing. Simultaneously, GATA6 targets closed chromatin and initiates extensive remodeling culminating in the establishment of fragile nucleosomes flanked by ordered nucleosome arrays and increased accessibility. This is enhanced by rapidly expressed PrE TFs (SOX17) and by pluripotency TFs repurposed for differentiation (OCT4/SOX2). Accordingly, depletion of OCT4 during GATA6 induction decreases *Gata6* expression, alters GATA6 and SOX17 binding and impairs differentiation. Therefore, pioneer TFs orchestrate complex regulatory networks involving many if not all available pioneer TFs, including those required to support the original identity of differentiating cells.

The acquisition of new cell identities during differentiation processes requires the activation of genes and their regulatory elements[1] even when they are tightly packed into chromatin[2] and inaccessible to many transcription factors (TFs)[3]. Indeed, the nucleosome, the basic structural unit of the chromatin, constituted of ~146 bases of DNA wrapped around an octamer of histone proteins[4], represents a barrier for most TFs to bind[5]. Important exceptions exist, however, referred to as pioneer TFs that bind to their cognate motifs even when packed into nucleosomes[6]. By licensing otherwise closed and inactive regulatory elements, pioneer TFs initiate large reprogramming events associated with the establishment of the transcription profiles driving and assisting cell differentiation[7–9].

Members of the GATA family have long been shown to display properties of pioneer TFs[10]. GATA1 was first shown to bind in vitro to a nucleosome harboring binding motifs, leading to the destabilization of histone-DNA contacts and to increased sensitivity to digestion by nucleases[11]. Later, in the seminal work coining the term pioneer TF, another member with a recognized role in embryonic[10,12] and extra-embryonic endoderm differentiation[13,14], GATA4, was shown to bind and open nucleosomal arrays in vitro[15]. Subsequently, other members, including GATA6, were shown to bind nucleosome arrays in vitro or

[1]Department of Developmental and Stem Cell Biology, Institut Pasteur, Université Paris Cité, CNRS UMR3738, Epigenomics, Proliferation and the Identity of Cells Unit, Paris, France. [2]Department of Developmental and Stem Cell Biology, Institut Pasteur, Université Paris Cité, CNRS UMR3738, Early Mammalian Development and Stem Cell Biology Group, Paris, France. ✉e-mail: michel.cohen-tannoudji@pasteur.fr; pnavarro@pasteur.fr

suggested to do so in cellular contexts where they trigger or anticipate chromatin opening[16–23]. Thus, GATA factors often occupy a top position within the hierarchies driving differentiation, especially within endoderm lineages. A paradigmatic example is provided by GATA6, which drives the early embryonic segregation of the Primitive Endoderm (PrE) that will lead to extra-embryonic tissues such as the yolk sac, from the pluripotent epiblast–the source of somatic tissues[24].

Rapidly recognized as an essential regulator of the PrE and its first derivatives[25], GATA6 was shown to drive the differentiation of mouse embryonic stem (ES) cells, derived from the undifferentiated epiblast, into PrE-like cells[14]. In vivo, GATA6 was shown to first mark, and then trigger PrE differentiation, via a temporally-ordered circuitry whereby it activates SOX17, a High Mobility Group pioneer TF[26], GATA4 and SOX7, all contributing to efficient PrE differentiation[13,27–32]. However, although acting as the upstream regulator in this circuit, GATA6 is unlikely to act alone. Indeed, in line with work suggesting that pluripotency TFs (PTFs) are generally involved in differentiation[33], numerous studies indicate that they contribute to PrE differentiation, suggesting a functional interaction with GATA6, the nature of which and the potential mutual dependencies are not understood. These include SOX2 or NANOG, shown to bind with GATA6 to preserve plasticity and prime PrE differentiation in early differentiating ES cells[22,34]; OCT4, a recognized pioneer TF[35] that is required for PrE differentiation in vivo[36–38] and promotes PrE differentiation upon its overexpression in ES cells[39] and its interaction with SOX17[40]; ESRRB, capable of binding nucleosomal DNA in vitro[41], this nuclear receptor first primes[42,43] and then promotes PrE differentiation[44,45] upon the loss of NANOG[43].

Overall, several lines of evidence indicate that the differentiation into PrE requires both the activity of GATA6, potentially as a pioneer TF, and that of several other PrE and pluripotency regulators, which also display pioneering properties. Given the cooperation existing between the pioneer TFs TFAP2A and OCT4[46] to confer developmental competence to neural crest cells[47,48], and the cooption of somatic pioneer TFs by OCT4/SOX2 during cellular reprogramming[49], we aimed at longitudinally exploring how GATA6 induction in ES cells induces transcriptional changes in relation to its pioneering role and the activity of SOX17, OCT4, SOX2 and ESRRB. We found that they engage in a complex network of molecular and epistatic interactions to execute the multitude of gene regulatory tasks required for PrE differentiation, highlighting the large cooperativity that exists between available pioneer TFs during cell fate changes.

## Results

### Distinct dynamics of GATA6 binding are associated with distinct gene regulatory outcomes

To study how GATA6 triggers PrE differentiation, we generated ES cells expressing GATA6 at physiological levels upon exposure to doxycycline (Dox)[14,50]. Indeed, the expression of *Gata6* continuously increased upon Dox treatment, until reaching levels similar to those measured in XEN cells (Supplementary Fig. 1A), a stem cell population derived from early mouse embryos and reminiscent of the PrE[51]. Specifically monitoring endogenous *Gata6* expression revealed its fast activation, indicating the establishment of a positive autoregulatory loop (Supplementary Fig. 1B). GATA6-induced ES cells underwent drastic morphological changes over the course of several days (Supplementary Fig. 1C), accompanied by a progressive modification of marker gene expression from pluripotency (OCT4, SOX2, ESRRB) to PrE-specific proteins (GATA6, SOX17, PDGFRA), characterized by a substantial and transient overlap between the two categories of a priori opposing TFs (Supplementary Fig. 1C–E). Transcriptomic comparison of undifferentiated ES cells, GATA6-induced cells and XEN cells further confirmed the progressive nature of the conversion (Fig. 1A). We identified 6 clusters of genes up- or downregulated in successive waves (Supplementary Fig. 1F), initiated either rapidly after Dox

induction or at later stages (Fig. 1B and Source Data 1). We then interrogated whether these clusters were enriched in up- or downregulated genes in the PrE of the blastocyst[52]. We found that only genes deregulated at early stages displayed a global concordant regulation in the PrE (Fig. 1C). Notable examples of early down- and upregulated genes are PTFs such as *Nanog*, *Esrrb*, or *Tfcp2l1*, and known drivers of PrE differentiation such as *Sox17*, *Gata4*, or *Sox7*, respectively (Source Data 1). We conclude that GATA6 elicits a fast and PrE-specific gene response that rapidly down-regulates pluripotency genes and simultaneously up-regulates PrE genes, including *Gata6* itself. To test whether the observed gene responses are linked to GATA6, we performed genome-wide localization studies. We observed global changes in GATA6 binding from 8 h of induction to 4 days, with a progressive acquisition of a binding profile similar to that observed in XEN cells (Fig. 1D). We identified 3 global behaviors of GATA6 (Source Data 1), easily observed in single loci (Supplementary Fig. 1G, H and Fig. 1E): regions rapidly targeted by GATA6, which can either maintain (thereafter Early&Constant sites) or instead subsequently lose binding as PrE differentiation advances (thereafter Transient sites), or sites that are bound by GATA6 only once differentiation has substantially progressed (thereafter Late sites). To provide support to these dynamics, we compared the binding sites reported here with a fully independent dataset of GATA6 binding in a related population of naïve PrE cells (nEnd[34]). As expected, we found a large overlap with Early&Constant and Late clusters only (Fig. 1F). Upon statistical confrontation[53] of the 3 GATA6 clusters to the 6 groups of differentially expressed genes, we revealed a very clear relationship, with Early&Constant sites being strongly enriched near genes that are rapidly activated (Fig. 1G, left panel) and Late sites being associated with genes that are activated later (Fig. 1G, middle panel). In contrast, Transient sites were enriched around genes that are downregulated early or at mid-stages of the differentiation induced by GATA6 (Fig. 1G, right panel). Thus, the temporal and dynamic aspects of GATA6 binding are associated with the kinetics (early/late) and modalities (up/down) of gene expression changes driving PrE differentiation.

### Chromatin accessibility and DNA binding motifs underlie GATA6 binding dynamics

We then aimed at identifying the molecular properties that could underlie differential recruitment kinetics and stability of GATA6 binding. We reasoned that the first binding events could be associated with pre-existing chromatin accessibility. To test this, we ranked all GATA6 binding sites by decreasing levels of accessibility in undifferentiated ES cells (Source Data 1) and found a strong correlation with the binding levels observed at early time-points, especially after 8 h of Dox induction, for each group of GATA6 binding regions (Fig. 2A). Moreover, Late regions were those displaying the lowest proportion of already accessible sites and, conversely, Early&Constant and more especially Transient sites displayed a higher proportion (Fig. 2A). Even though the enrichment of accessible sites over Transient regions is statistically significant, as is their depletion over Late sites (Supplementary Fig. 2A–C top panels), which are also enriched for heterochromatin marks (Supplementary Fig. 2A–C bottom panels and Source Data 1), these relationships are not absolute. Thus, additional parameters may contribute to GATA6 dynamics. Accordingly, we found that Early&Constant and Late regions displayed the highest and lowest density of high-quality GATA6 DNA motifs, respectively (Fig. 2B, Supplementary Fig. 2A–C and Source Data 1). At Transient sites, which display an intermediary average occurrence of GATA6 motifs, as well as at a small proportion of Late sites, we observed a negatively correlated presence of good GATA6 motifs with chromatin accessibility: the less the regions are accessible, the highest is the density and the quality of GATA6 motifs (Fig. 2B). This suggests that a combination of pre-existing accessibility and the occurrence of good GATA6 motifs drive, at least partially, GATA6 binding dynamics, in such a way that at

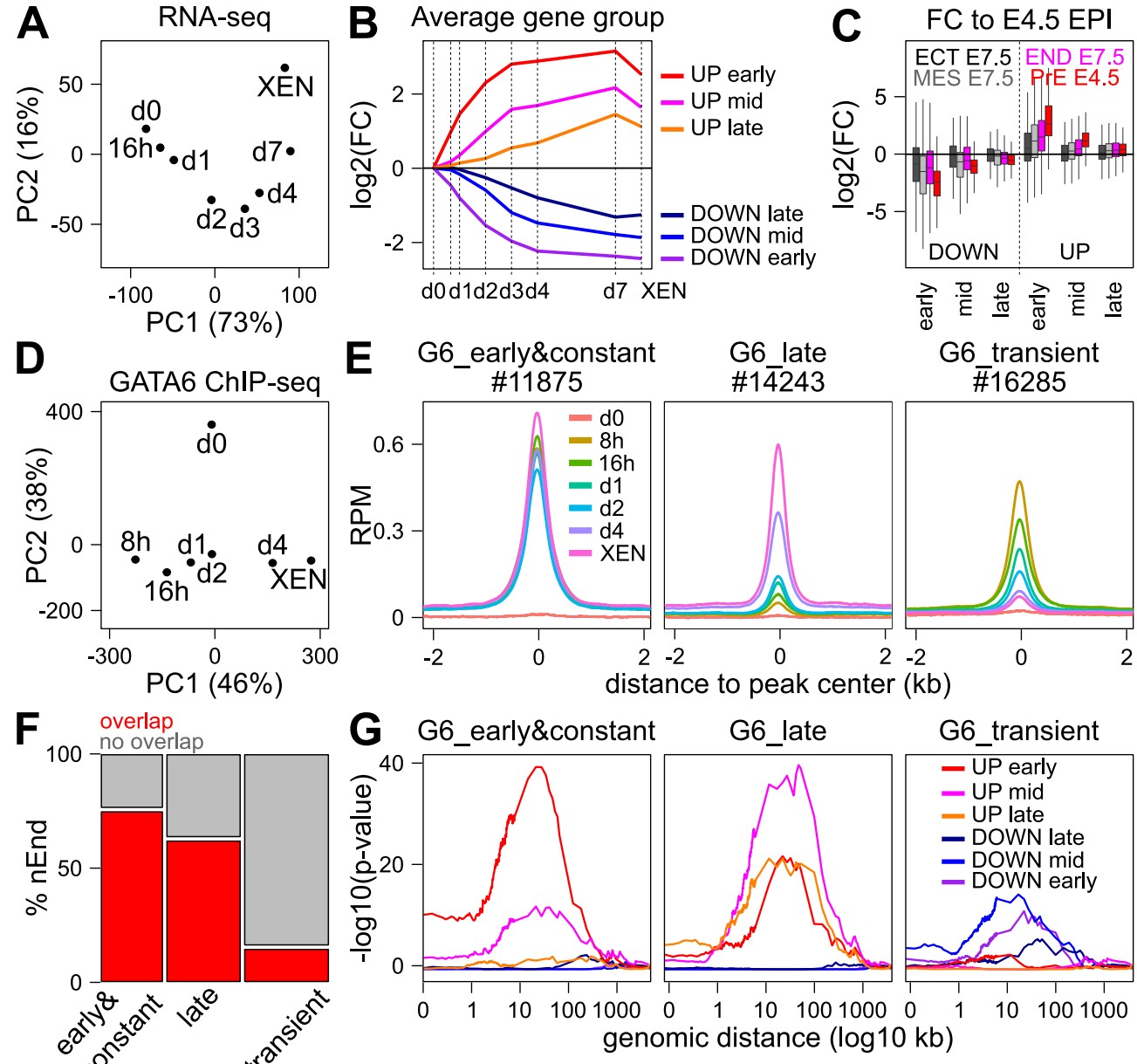

**Fig. 1 | Different dynamics of GATA6-binding induce successive waves of gene expression changes. A** Principal component analysis (PCA) of differentially expressed genes across all analyzed time-points and in XEN cells. **B** Average expression profile of 6 distinct clusters of early, mid or late up- or downregulated genes. Early, Mid and Late responding genes were selected as those with expression changes observed from 16 or 24 h onwards (Early), from 2 or 3 days onwards (Mid) or after day 4 or 7 (Late; see Supplementary Methods for details). **C** Boxplots (median−bar; 25-75% percentiles−box; 1.5-folds the inter-quartile range−whiskers) of the log2 fold-change of the 6 gene clusters shown in (**B**) between the E7.5 ectoderm, endoderm and mesoderm (ECT, END and MES, respectively), or the E4.5 PrE, compared to the E.4.5 epiblast. **D** PCA of GATA6 binding regions identified across time-points and in XEN cells. **E** Average GATA6 enrichment profiles expressed in reads per millions (RPM) across time-points and in XEN cells for 3 clusters obtained by k-means clustering of log2 input normalized counts, displaying different temporal binding dynamics, centered on the GATA6 summit (see Methods for details). **F** Proportion of GATA6 binding regions in each category shown in (**E**) that overlap with binding regions identified in naïve Endoderm cells (nEnd). The statistical significance of the enrichment observed at Early and Late sites was tested with two-sided Chi-square followed by Fisher Exact tests ($p < 2.2e$-16). **G** Statistical association between each GATA6 binding cluster and the 6 groups of differentially expressed genes shown in (**B**), measured with Fisher Exact tests at increasing distances. Source data are provided as a Source Data file.

highly accessible regions, motifs of less quality are sufficient to trigger fast GATA6 binding and, in reverse, at regions with closed chromatin very good motifs are needed. Inspection of other motifs showed that Transient sites were characterized by a low presence of the PrE TF SOX17 motif and an enrichment of good motifs for PTFs OCT4/SOX2 and ESRRB (Source Data 1), especially over accessible regions (Fig. 2B) bound by PTFs before differentiation (Fig. 2C). Conversely, Late sites displayed the opposite pattern (Fig. 2B and Supplementary Fig. 2A−C) with high quality SOX17 motifs (Fig. 2B) and an almost complete lack of

PTFs motifs (Fig. 2B) and binding (Fig. 2C). Therefore, different combinations of TF motifs likely contribute to differential GATA6 binding dynamics: Early&Constant regions are strongly enriched for good GATA6 and SOX17 motifs; Late regions are more strongly associated with SOX17 motifs only; Transient GATA6 binding regions are characterized by the presence of PTFs motifs (Fig. 2B), coinciding with extensive PTFs binding (Fig. 2C) and chromatin accessibility (Fig. 2A). In support, both DNA motifs and chromatin states, particularly GATA6 motifs and chromatin accessibility, were found as good predictors of

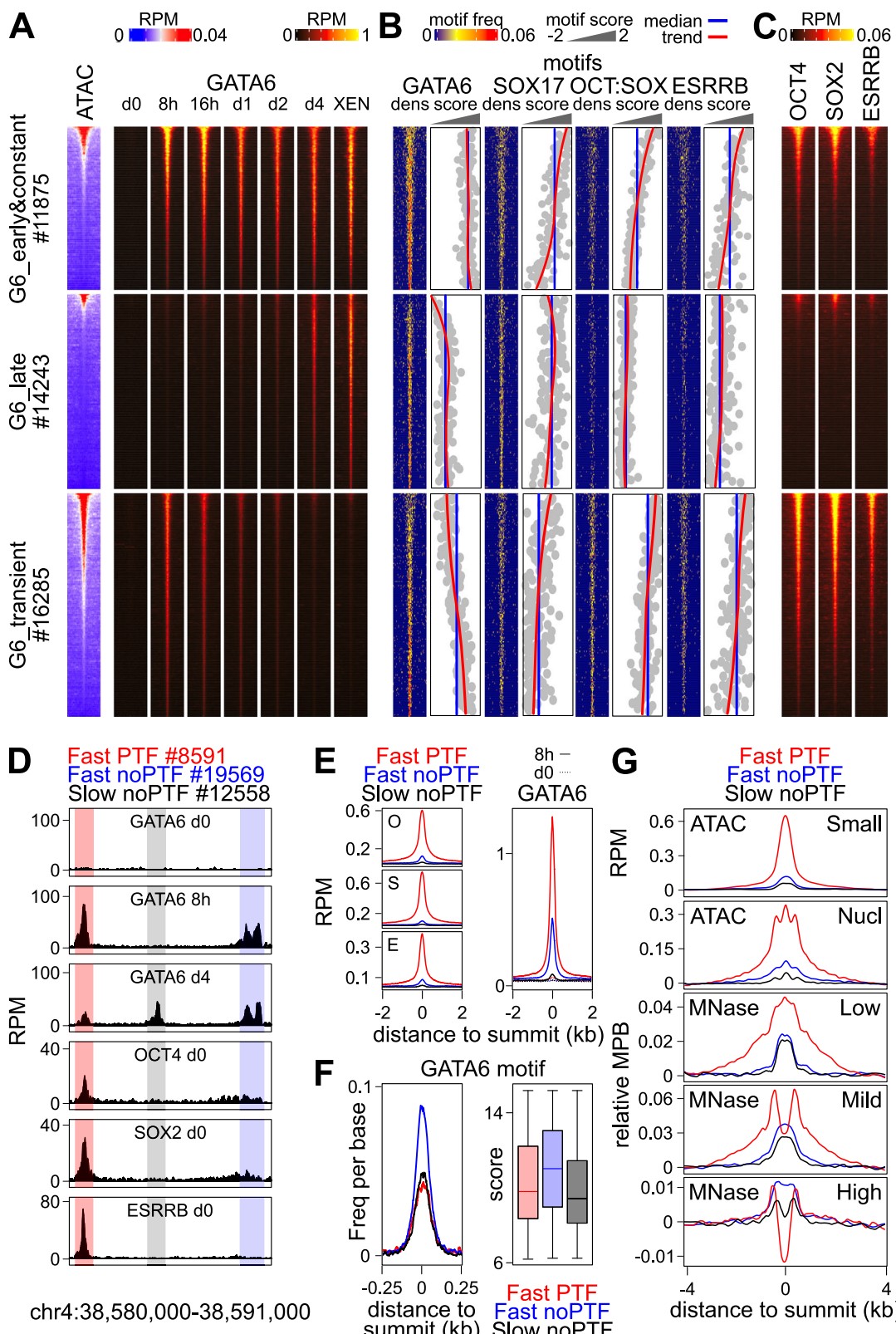

the classification of GATA6 binding regions using machine learning (Supplementary Fig. 2D)[54].

## Pluripotency TF-binding sites harboring fragile nucleosomes are robustly targeted by GATA6

To more directly explore the relationships between PTFs, chromatin accessibility and the dynamics of GATA6 binding, we combined Early&Constant and Transient regions, which display fast GATA6 recruitment, and divided them as associated with PTFs binding or not (Fig. 2D). For comparison purposes we used Late GATA6 regions, which recruit GATA6 very slowly and generally lack PTFs binding (Fig. 2A, D). We found that regions of fast GATA6 recruitment harboring PTFs binding in ES cells (Fig. 2E, left panels) showed the highest levels of GATA6 binding at 8 h (Fig. 2E right panel), even though these

**Fig. 2 | Pre-existing chromatin accessibility, DNA binding motifs and pluripotency TF binding, underlie GATA6 binding dynamics. A** Heatmaps depicting chromatin accessibility (ATAC-seq) and GATA6 enrichments, over 4kb-long regions centered on GATA6 summits, ranked by decreasing accessibility in ES cells at each GATA6 cluster. **B** Occurrence and quality (identity score to the consensus of the best motif of each region, z-scored to scale between different motifs) of DNA binding motifs across the regions shown in (**A**), in the same order as in (**A**). Within each score panel is shown the global median of all regions per cluster (blue) and a trend line (red) of the data (gray). **C** Analysis of pluripotency TF binding (reads per million; RPM) across the 3 GATA6 clusters, presented and ordered as in (**A**). **D** Example of different behaviors of GATA6 and pluripotency TFs (PTF). **E** Left; average profile of OCT4 (O), SOX2 (S) and ESRRB (E) at the 3 groups of regions

illustrated in (**D**). Right; binding of GATA6 across the same clusters. All four plots show the average profile (RPM) over 4 kb centered on the GATA6 summit.
**F** Distribution of GATA6 motifs around the GATA6 summit (left panel), together with the motif score of the best motif present in each region (right panel, median−bar; 25–75% percentiles−box; 1.5-folds the inter-quartile range−whiskers).
**G** Average profiles of chromatin accessibility (small ATAC-seq fragments) and nucleosomes (measured by long fragments detected by ATAC-seq or by MNase-seq using different concentrations of MNase−Low, Mild, High). ATAC-seq is expressed in RPM and MNase-seq in normalized midpoints per billion reads (MPB; see Methods for details). All plots show the average profile over 8 kb centered on the GATA6 summit. Source data are provided as a Source Data file.

are not the regions with the highest density nor the best quality of GATA6 motifs (Fig. 2F). Indeed, the regions displaying the best motif composition are those recruiting GATA6 rapidly, albeit at more modest levels, in the absence of PTFs binding (Fig. 2E, F). As expected, regions bound by PTFs were highly accessible in ES cells compared to those lacking their binding (Fig. 2G, top panel), suggesting that their binding might be associated with reduced nucleosomal occupancy. However, when we profiled nucleosomes in the same dataset (Fig. 2G, second top panel), the only observed difference between the three groups was the amplitude of the signal, which represents differential accessibility rather than specific positioning of the nucleosomes. In fact, all three regions show the presence of 3 nucleosomes, including a central prominent one located right on the GATA6 binding summit (Fig. 2G, second top panel). While this is in accord with the pioneering nature of GATA factors[15,16,23,55], the simultaneous detection of accessibility and nucleosomes suggests that they are fragile[56]. To test this, we explored previous data generated using increasing concentrations of Micrococcal Nuclease (MNase) to map nucleosomes with different stability in ES cells[57]. We observed that regions binding PTFs display a strong nucleosomal signal at the GATA6 summit exclusively when low MNase conditions are used (Fig. 2G; Low), with nucleosome signal progressively decaying with increased MNase concentrations (Fig. 2G; Mild) until it completely disappears at the highest MNase digestion conditions, leading to a profound and apparent nucleosome-depleted region (Fig. 2G; High). In contrast, in regions lacking PTFs binding the observed nucleosome profiles were less sensitive to MNase digestion conditions (Fig. 2G). Since these regions are not accessible, we conclude that they are characterized by more stable nucleosomes. Thus, PTFs fragilize nucleosomes, leading to a net advantage for fast and robust GATA6 binding even at regions harboring GATA6 motifs of relatively low quality.

## SOX17 recruitment supports long-term GATA6 binding and extensive nucleosome remodeling

Next, we sought to investigate the evolution of chromatin accessibility at Early&Constant, Late and Transient GATA6 binding regions (Source Data 1). We found that at already open Early&Constant and Late sites chromatin accessibility was maintained throughout differentiation, whereas at Transient sites it was eventually lost by day 4 of GATA6 induction (Fig. 3A). At the remaining and initially closed Early&Constant and Late sites, but not at Transient sites, a global strong increase was instated (Fig. 3A), rapidly at Early&Constant sites (by days 1/2) and more slowly at Late sites (by days 2/4). However, these dynamics were relatively delayed compared to the timing of GATA6 binding over these groups, especially at Early&Constant sites that bind GATA6 as early as 8 h following its induction (Figs. 1, 2). Given the prevalence of SOX17 DNA motifs over Early&Constant and Late regions compared to Transient sites (Fig. 2B), and the fact that GATA6 binds rapidly at the *Sox17* locus to activate its expression (Supplementary Fig. 3A, B and Supplementary Fig. 1E), we reasoned that it could contribute to chromatin opening of these sites. Hence, we profiled SOX17 binding and found that it is efficiently recruited to a large fraction of GATA6 bound regions

(Fig. 3B, Supplementary Fig. 3B, C and Source Data 1), predominantly at Early&Constant and Late sites and only anecdotally at Transient sites (Fig. 3B, Supplementary Fig. 3C), matching the distribution of its DNA motif (Fig. 2B, Supplementary Fig. 2). Moreover, comparing the average trends of chromatin accessibility, GATA6 and SOX17 recruitment, we observed that SOX17 is recruited concomitantly to, or slightly before, any measurable increase in chromatin accessibility, whereas GATA6 binding occurs earlier (Supplementary Fig. 3D). The association of SOX17 binding with faster and stronger chromatin opening was confirmed by dividing GATA6 bound regions in those recruiting SOX17 or not, since we observed that only the former display rapid, strong and constant GATA6 binding and chromatin accessibility (Fig. 3C, top). Moreover, SOX17-bound regions were also subject to a prominent reorganization of nucleosomes, as established by MNase-seq using sufficiently drastic conditions to observe the disappearance of fragile nucleosomes (Fig. 3C, bottom). Indeed, an apparent central nucleosome-depleted region flanked by ordered nucleosomes appeared as early as 8 h post-induction and consolidated throughout the 4 days of PrE differentiation at SOX17-bound loci only (Fig. 3C, bottom). Therefore, SOX17, an early target of GATA6, reinforces GATA6 binding and strongly contributes to chromatin opening and nucleosome organization. Motivated by these observations, and with the aim of avoiding confounding variables due to the presence of distinct initial states driven by PTFs (Fig. 2), we further stratified GATA6 binding regions by their dynamics (Early&Constant, Late, Transient), by the presence of PTFs before differentiation and by the recruitment of SOX17 during differentiation, excluding poorly represented combinations (Supplementary Fig. 3C, Source Data 1). This analysis confirmed the fastest and strongest binding of GATA6 (Supplementary Fig. 4A) as well as the acquisition of chromatin accessibility (Supplementary Fig. 4B) at regions binding PTF before differentiation and recruiting SOX17. Moreover, for all categories the recruitment of SOX17 appeared as an amplifier of chromatin accessibility, especially after d1 (Supplementary Fig. 4B). Further, comparing nucleosome profiles derived from accessibility datasets (Supplementary Fig. 4C) and from stringent MNase digestion (Supplementary Fig. 4D), we could confirm the appearance of fragile nucleosomes at GATA6 binding regions during GATA6 induction, especially at those sites recruiting SOX17, more rapidly at Early&Constant than at Late sites and only transiently at Transient sites (Supplementary Fig. 4C, D). Notably, by day 4 of differentiation, the initial differences mediated by PTFs had been fully erased. Transient sites showed no signs of fragile nucleosomes (in accord with reduced levels of accessibility), and Early&Constant and Late sites showed precisely ordered nucleosomal arrays flanking an apparent nucleosome-depleted region at the GATA6 summit. Summarizing these complex dynamic patterns by assessing nucleosome order (Fig. 3D, top) and nucleosome density at the GATA6 binding summit (Fig. 3D, bottom) across GATA6 binding groups provided quantitative support to the conclusion that GATA6 initiates, and SOX17 drastically enforces, an extensive nucleosome remodeling that leads to properly ordered nucleosomes around a central fragile nucleosome sitting at the GATA6 binding summit.

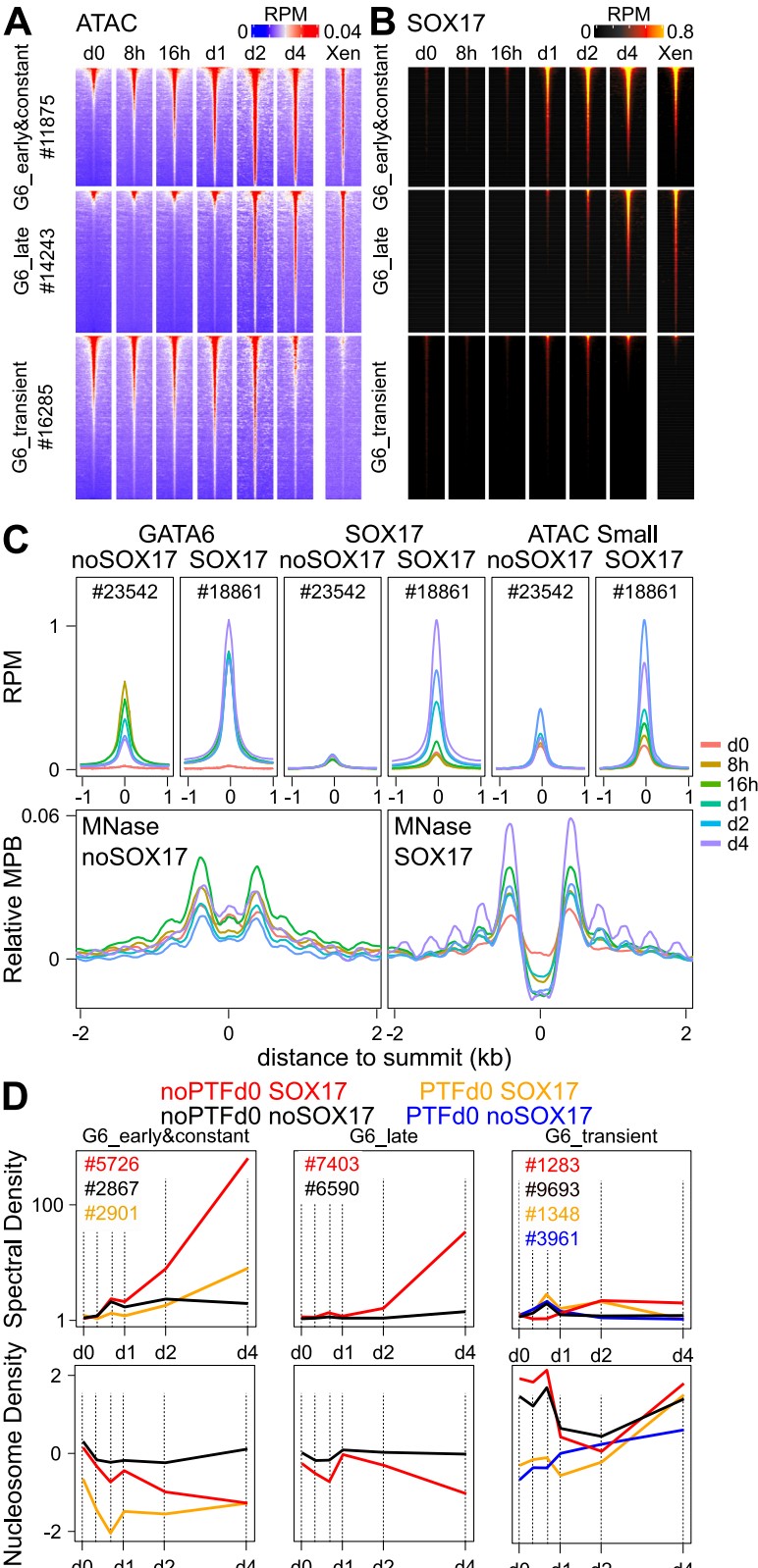

**Fig. 3 | SOX17 recruitment promotes and stabilizes GATA6 binding, leading to increased chromatin remodeling. A** Evolution of chromatin accessibility measured by ATAC-seq over GATA6 binding groups. Presented and ordered as in Fig. 2A. **B** Evolution of SOX17 binding across the same regions, ordered as in A and Fig. 2. **C** Average profiles of GATA6 binding, SOX17 binding, chromatin accessibility (small ATAC-seq fragments) and nucleosome positioning (MNase-seq) at regions displaying or not SOX17 recruitment, throughout GATA6 induction (colored lines). All plots are centered on the GATA6 summit. **D** Top. Quantification of nucleosome order using the spectral density calculated from the profiles shown in Supplementary Fig. 4D at a period of 180 bases−nucleosome (150 bases) plus linker DNA (30 bases)−throughout GATA6 induction time-points. Bottom. Quantification of nucleosome density over 50 bp centered on the GATA6 summit calculated from the profiles shown in Supplementary Fig. 4D. Source data are provided as a Source Data file.

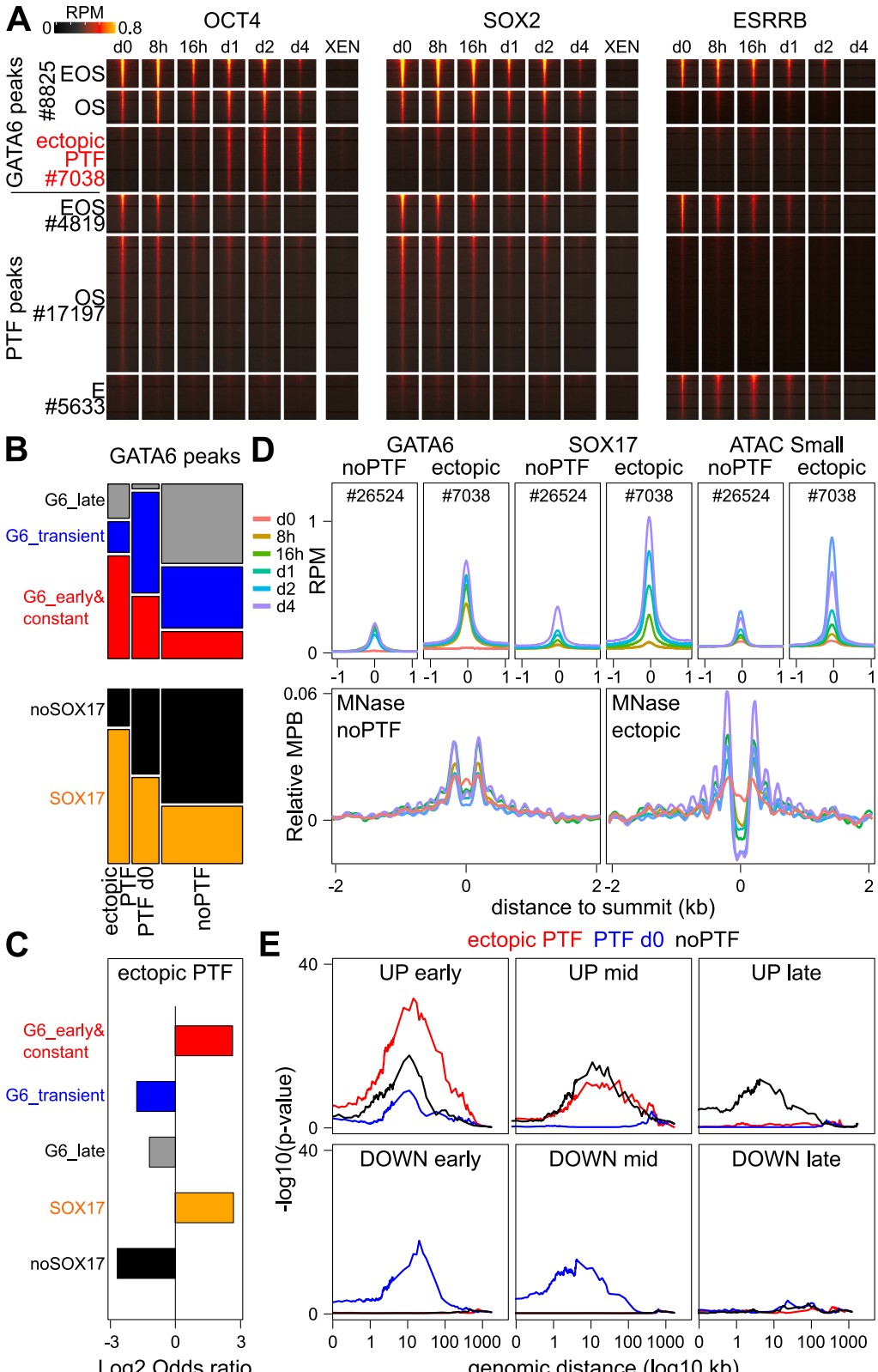

## OCT4/SOX2 promote nucleosome remodeling at GATA6/SOX17 targets and PrE gene activation

We next aimed at monitoring PTFs binding along GATA6 induction, with the initial idea of following how the pluripotency network is dismantled. Upon OCT4, SOX2 and ESRRB profiling (Fig. 4A, Supplementary Fig. 5A and Source Data 1), we observed that GATA6 binds consistently to around a fifth of all sites bound by PTFs in undifferentiated ES cells (Fig. 4A). Moreover, GATA6 preferentially targets regions characterized by high levels of PTF binding, especially OCT4 (Fig. 4A and Supplementary Fig. 5B). In contrast to our expectations, we found that at regions targeted by GATA6, binding of PTFs was detected during a longer period after GATA6 induction than at regions not targeted by GATA6, which lose almost all signs of PTFs binding during the very first days of GATA6 induction (Fig. 4A and

**Fig. 4 | GATA6 maintains and promotes ectopic binding of pluripotency TFs during PrE differentiation. A** Binding dynamics of OCT4, SOX2 and ESRRB upon GATA6 induction, both at GATA6 peaks showing pluripotency TF binding before or during differentiation (i.e., GATA6 peaks lacking pluripotency TF binding are omitted) and at other pluripotency TF binding sites in undifferentiated ES cells. **B** Correlations between categories of GATA6 binding segregated by pluripotency TF binding ($X$ axis) and by GATA6 dynamics (top; $Y$ axis) or by SOX17 binding (bottom; $Y$ axis). The dependency between variables was assessed with two-sided Chi-square tests ($p < 2.2e\text{-}16$). **C** Log2 odds ratio of the proportion of regions displaying ectopic pluripotency TF binding across different GATA6 binding groups shown on the left. The statistical significance of the enrichments/depletions was assessed with two-sided Fisher's Exact tests ($p < 1.3e\text{-}155$). **D** Average profiles of GATA6 binding, SOX17 binding, chromatin accessibility (small ATAC-seq fragments) and nucleosome positioning (MNase-seq) at regions displaying no pluripotency TF binding at all (noPTF) or an ectopic recruitment (ectopic), throughout GATA6 induction (colored lines). All plots are centered on the GATA6 summit. **E** Statistical association of GATA6 binding regions divided by pluripotency TF binding groups, with the six groups of differentially expressed genes shown in Fig. 1B, measured with two-sided Fisher Exact tests at increasing distances. Source data are provided as a Source Data file.

Supplementary Fig. 5B). While a global reduction of PTFs binding is expected given the downregulation of their expression (Supplementary Fig. 1D, E and Source Data 1), their transient retention at GATA6 bound regions suggests that the declining pool of remaining molecules is specifically redirected at these sites. In addition, our analyses identified a cluster of GATA6 regions experiencing ectopic binding of PTFs during differentiation, especially of OCT4/SOX2 (Fig. 4A and Supplementary Fig. 5A, B). These regions were found enriched for Early&Constant GATA6 binding regions and for SOX17 recruitment (Fig. 4B, C), displayed high levels of GATA6 and SOX17 binding and rapidly gained accessibility (Fig. 4D top) to levels even higher than those observed at the best PTFs sites in ES cells (Supplementary Fig. 5C, top row). In contrast, PTFs' binding sites not targeted by GATA6 progressively lost chromatin accessibility (Supplementary Fig. 5C, top row). Analysis of nucleosome profiles from accessibility datasets confirmed previous observations, with the detection of nucleosomes at the GATA6 summit (Supplementary Fig. 5C middle row). Using stringent MNase digestion, we could, however, unmask the appearance of an apparent nucleosome-depleted region concomitantly with ordered flanking nucleosomes (Fig. 4D bottom, Supplementary Fig. 5C bottom row), which was much more pronounced at regions of ectopic PTFs binding than at those not associated with their recruitment. Similarly, focusing on regions bound by PTFs in undifferentiated ES cells and comparing those that are either targeted or not by GATA6, we could confirm that only the later exhibit a progressive invasion of the region by stable nucleosomes, whereas those associated with GATA6 preserve an apparent nucleosome depleted region that represent the presence of fragile nucleosomes, particularly at early time-points after Dox induction (Supplementary Fig. 5C bottom row). Furthermore, we found that GATA6 sites with ectopic OCT4/SOX2 binding were strongly and very specifically associated with gene activation events occurring early following GATA6 induction, whereas regions not associated with PTFs were equally associated with gene activation at any time (Fig. 4E). In contrast, the regions displaying PTFs before differentiation tend to be associated with early repressive gene changes (Fig. 4E). We conclude that GATA6 and SOX17 maintain and repurpose PTFs binding, leading to a prominent nucleosome remodeling and the rapid activation of PrE genes.

## OCT4 promotes GATA6 and SOX17 activity to activate PrE-associated genes

Given the strong link existing between ectopic OCT4/SOX2 binding, strong GATA6 recruitment and nucleosome remodeling and the activation of early GATA6-responsive genes, we decided to test whether OCT4 is strictly required for PrE differentiation. To address this, we triggered siRNA-mediated *Oct4* knockdown concomitantly with Dox induction of GATA6. After 2 days, when major ectopic OCT4 recruitment is observed (Fig. 4), we observed a consistent reduction of *Oct4* mRNA expression (Supplementary Fig. 6A) that led to a nearly full abrogation of OCT4 by day 4 (Supplementary Fig. 6B, top and Source Data 2), when control cells subject to non-targeting siRNAs still displayed low but measurable OCT4 levels (Supplementary Fig. 6B, top and Source Data 2). This accelerated loss of OCT4 was found to be accompanied by a lower efficiency of PrE differentiation: *Oct4*-kd cells exhibited a lack of clearly defined islands of PrE-like cells (Fig. 5A), a subpopulation of SOX17 negative cells (Fig. 5B, Supplementary Fig. 6B, middle and Source Data 2), and generally reduced PDGFRa expression (Fig. 5B, Supplementary Fig. 6B, bottom and Source Data 2). We conclude that the experimental reduction of OCT4 expression during GATA6 induction decreases the efficiency of GATA6-mediated PrE differentiation. RNA-seq analyses further confirmed this conclusion, as we found 485 genes that were consistently downregulated at day 2 of simultaneous GATA6 induction and *Oct4* knockdown, and 273 that were upregulated (Supplementary Fig. 6C, Source Data 1). Statistical analyses showed a strong link between OCT4 and GATA6-responsive genes, with genes downregulated upon *Oct4* knock-down being strongly enriched in genes rapidly upregulated by GATA6 (Supplementary Fig. 6D). Moreover, while GATA6-induced genes are generally associated with PrE differentiation in vivo (Fig. 1C), those that are also downregulated upon *Oct4* knock-down displayed stronger and more homogeneous induction levels in the E4.5 embryonic PrE versus the undifferentiated epiblast (Supplementary Fig. 6E). Thus, OCT4 promotes the activation of a substantial fraction of GATA6 targets, improving PrE differentiation. To investigate how OCT4 promotes GATA6 activity, we profiled GATA6 and SOX17 after 2 days of simultaneous GATA6 induction and *Oct4* knockdown. We found that among our collection of GATA6 peaks, 2651 regions displayed lower GATA6 and SOX17 binding, whereas 3636 showed higher levels upon *Oct4* knockdown (Fig. 5C and Source Data 1). Notably, a strong correlation could be observed between the effects triggered upon *Oct4* knock-down and the presence of PTF binding, with a strong enrichment for regions bound by PTFs at sites displaying reduced GATA6/SOX17 binding (Fig. 5D). Conversely, an enrichment for regions lacking PTFs binding could be noted for those displaying increased GATA6/SOX17 binding (Fig. 5D). Therefore, we conclude that the premature loss of OCT4 leads to inefficient GATA6 and SOX17 recruitment at a subset of GATA6 targets where OCT4 is needed to maximize the binding and activity of TFs driving PrE differentiation. In turn, other sites not associated with OCT4 experience increased binding activity. To link these effects with gene expression changes, we established the enrichment profile of OCT4-responsive genes in the vicinity of the regions displaying differential GATA6/SOX17 binding (Fig. 5E). We observed that genes downregulated upon *Oct4* knock-down were strongly enriched close to regions subject to a loss of GATA6/SOX17 binding, whereas upregulated genes were enriched, albeit at lower levels, in proximity of regions displaying increased binding. This strong correlation highlights the functional link existing between gene expression changes and variations in GATA6/SOX17 binding, especially regarding genes activated by GATA6/SOX17 that require OCT4 to reach maximal expression. Accordingly, when we monitored the expression fold change of the closest genes of each of the regions displaying altered GATA6/SOX17 binding, we observed a clear downregulation tendency for the closest genes to the regions losing GATA6/SOX17 upon Oct4 knock-down (Supplementary Fig. 6F). We conclude that OCT4 promotes GATA6 and SOX17 binding and activity at regions controlling the expression of a selected group of PrE-associated genes.

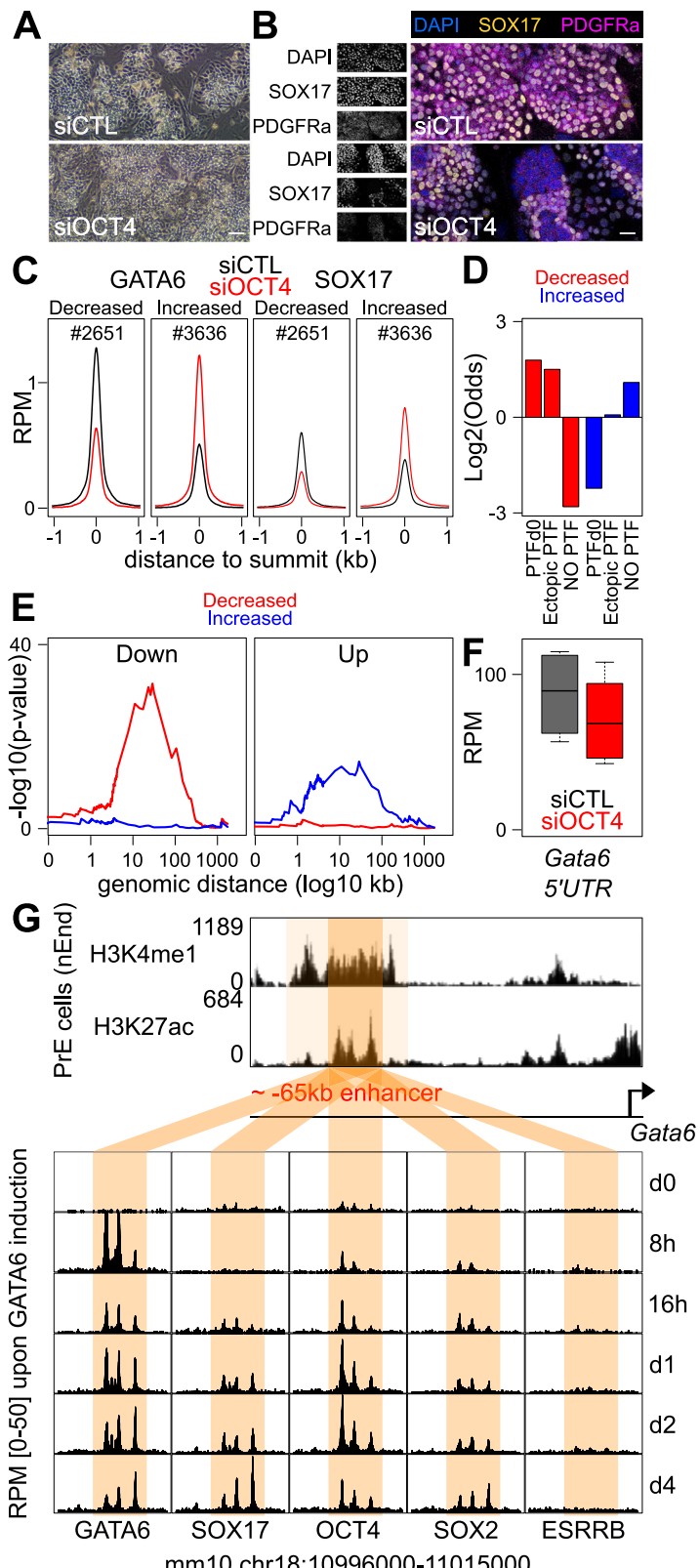

## OCT4 induces *Gata6* expression and is essential for PrE differentiation

Among the PrE-associated genes activated by OCT4 (Source Data 1), we noticed *Gata6* itself, which we measured by focusing on its 5'UTR, exclusively present in the endogenous alleles and not in the ectopic transgene (Fig. 5F). Exploration of our TF binding data around the *Gata6* locus (Fig. 5G) identified a region of fast GATA6 and SOX17 binding, where OCT4/SOX2 are ectopically recruited upon GATA6 induction and, notably, which displays the chromatin signature of an active enhancer, as observed using published datasets[34]. Upon *Oct4* knockdown, we further observed reduced GATA6 binding at the *Gata6* enhancer, suggesting a direct effect of OCT4 during PrE differentiation to promote *Gata6* expression (Supplementary Fig. 6G). Thus, the use of an ectopic GATA6 induction system may mask the true effects of

**Fig. 5 | OCT4 promotes GATA6 activity and *Gata6* expression. A** Representative photomicrographs of cells subject to GATA6 induction and to either non-targeting control siRNAs or *Oct4* knockdown for 4 days ($n = 6$ independent kd). The scale bar represents 100 μm. **B** Immunostaining of SOX17 and PDGFRa after 4 days of GATA6 induction and control or *Oct4* siRNAs (representative of 6 independent kd). The scale bar represents 30 μm. **C** Average profile of GATA6 and SOX17 binding at regions identified as showing decreased or increased GATA6 binding (FDR < 0.1 and absolute Fold-change > 0.5) after 2 days of GATA6 induction and *Oct4* knockdown. **D** Log2 odds ratio of the proportion of regions displaying decreased or increased GATA6 binding across groups of regions displaying different behaviors of pluripotency TF binding, shown on the bottom. The statistical significance of the enrichments/depletions was assessed with two-sided Fisher Exact tests ($p = 3.8e-$ 174 and 8.5e-107 for regions binding pluripotency TFs from d0 or ectopically, respectively). **E** Statistical association of GATA6 binding regions showing increased or decreased GATA6 binding upon *Oct4* knock-down, with genes displaying down or upregulated expression, measured with Fisher's Exact tests at increasing distances. **F** Expression of endogenous *Gata6* expression, measured by quantification of its 5′UTR absent in the ectopic transgene, upon control or *Oct4* knock-down during GATA6 induction (day 2; paired two-sided *T* test $p = 0.003992$; $n = 6$ independent kd; median−bar; 25–75% percentiles−box; 1.5-folds the inter-quartile range−whiskers). **G** Identification of an active enhancer based on histone marks (top) ~65 kb upstream of *Gata6,* showing no TF binding before differentiation but rapidly recruiting GATA6 and SOX17 and displaying ectopic OCT4/SOX2 binding during GATA6 induction. Source data are provided as a Source Data file.

OCT4, which may not only be required to maximize GATA6 activity but also its expression. To test this, we resorted to an established chemical induction of PrE differentiation[58,59] and used pluripotent cells that express OCT4 fused to an Auxin inducible degradation domain enabling fast and acute depletion upon IAA treatment[60,61]. While both IAA-untreated cells (Fig. 6A–C) and wild-type cells treated with IAA (Fig. 6C) efficiently differentiated into PrE, depleting OCT4 at different time points had strong and variable consequences. When OCT4 depletion was triggered concomitantly with the onset or at day 1 of differentiation, we observed a drastic mortality of the cells, uncapable of surviving the chemical induction (Fig. 6A). In contrast, when OCT4 depletion was performed at day 2 or later during the chemical induction, the differentiation into PrE was as efficient as in IAA-untreated cells (Fig. 6A–C). Hence, OCT4 is essential to allow early differentiating cells to survive, supporting the notion that OCT4 is strictly necessary for PrE differentiation. Furthermore, the quantification of PrE markers during the chemical induction of differentiation revealed low GATA6 and PDGFRA expression (Fig. 6B, C and Source Data 2) in the rare cells that succeeded in differentiating upon IAA treatment from day 1, but also in cells experiencing OCT4 depletion from day 2 and even day 3. The reduction of GATA6 expression, in line with that of *Gata6* upon *Oct4* knockdown, suggests that the early effects of OCT4 depletion observed upon chemical induction also involve the incapacity of the cells to properly induce GATA6 and initiate the large changes required for PrE differentiation. In contrast, once GATA6 has already irreversibly initiated differentiation, or when its expression is ectopically enforced, the depletion of OCT4 and the ensuing downregulation of endogenous *Gata6* are less detrimental.

### Integrated view of GATA6 roles during PrE differentiation

GATA6 leads to PrE differentiation via three major waves of gene regulation, with the earliest ones initiating the downregulation of the players of the pluripotency network and the upregulation of PrE genes, including SOX17. Through a meticulous stratification of GATA6 binding regions, based on GATA6 binding dynamics, the recruitment of SOX17 and the involvement of PTFs before or during differentiation, summarized in Supplementary Fig. 7, we can ascribe these regulatory waves to specific modalities of GATA6 action among its eight main categories (Fig. 7), supported by extensive remodeling of the gene regulatory network (see epistatic diagram in Fig. 7). Opportunistic GATA6 binding at regions made accessible by PTFs is fast even if GATA6 motifs are not particularly good, thanks to the fragilization of nucleosomes mediated by OCT4/SOX2/ESRRB. At these sites, good motifs for SOX17 and its subsequent recruitment determine if GATA6 binding will be stable or, on the contrary, transient. While transient GATA6 binding at these regions leads to their decommissioning and the dismantlement of pluripotency, stable GATA6/SOX17 binding leads to early PrE gene activation. GATA6 also binds at closed regions, acting as a pioneer TF with rapid or slow dynamics depending on the quality of GATA6 and SOX17 motifs. In the presence of good GATA6 motifs, its binding is fast and stable, but without the intervention of other TFs, it does not lead to either strong nucleosome remodeling or particular gene responses. In contrast, when GATA6 is accompanied by SOX17 and OCT4, transiently repurposed for differentiation, it leads to drastic nucleosome remodeling and early activation of PrE genes. When GATA6 motifs are of poorer quality, but provided that good SOX17 motifs are present, GATA6 and SOX17 are recruited later, leading to late gene activation. Thus, while GATA6 initiates a profound nucleosome remodeling characterized by the acquisition of central fragile nucleosomes flanked by ordered nucleosomal arrays and increased chromatin accessibility, this is fully maximized only upon binding of SOX17, OCT4 and SOX2. These TFs, which are regulated by GATA6, are essential for PrE differentiation and positively feed back on *Gata6* by reinforcing a *Gata6* positive autoregulatory loop. Thus, we have identified different modes of direct and indirect action of GATA6 to activate or repress transcription, acting either as an opportunistic or a pioneer TF and unfolding epistatic interactions to rewire the gene regulatory network.

## Discussion

This study adds to the broad literature supporting a role for GATA factors in pioneering processes[10,11,15–23]. However, we show here that GATA6 maximizes its activity in the context of the cooperative action of several pioneer TFs at defined regulatory elements. This notion might be particularly relevant for GATA factors, for which their pioneering activity is, to some extent, surprising, given that they have two consecutive zinc fingers that bind high-affinity motifs in free DNA but pose steric hindrance issues when DNA is bent in a nucleosome. For GATA3, like for other pioneer TFs binding partial motifs embedded in nucleosomes[26,35,55,62,63], this may be alleviated thanks to its nucleosomal binding to specific motif geometries consisting of repeated partial motifs[16]. However, we observe a strong dependence of GATA6 on its cognate motif, especially at inaccessible sites. In contrast to what has been suggested for SOX2[64], this is unlikely to depend on attaining especially high concentrations of GATA6 in our experimental system, which maintains physiological expression levels. Accordingly, GATA6 motifs were already found of better quality in nucleosomal than in free DNA in an independent study that did not use overexpression[55]. Strikingly, these good GATA6 motifs were present in regions also targeted by HNF3[55], which exhibits greater pioneering capacity than another GATA factor, GATA4[15]. This suggests that cooperative nucleosome attack by several pioneer TFs may help recognition and binding to canonical motifs even when embedded in nucleosomes. Indeed, GATA6 deploys proper pioneering activity in the presence of other pioneer TFs such as OCT4, SOX2 and SOX17. Thus, our data calls for a more nuanced view on the pioneering nature of GATA6, and, by extension, of other GATA factors. As proposed for SOX2[65], GATA6 acts both as a "settler" and opportunistic TF, when it binds at open regions and decommissions pluripotency enhancers, or as a "pilot" TF guiding further pioneer TF recruitment and remodeling activities (Fig. 7). Of course, we show that GATA6 binds at nucleosomal DNA, regardless of whether it is already made accessible by OCT4/SOX2 or if it is located in closed chromatin, be it naïve or heterochromatic. This, by itself, qualifies GATA6 as a pioneer TF; it remains true, however, that its

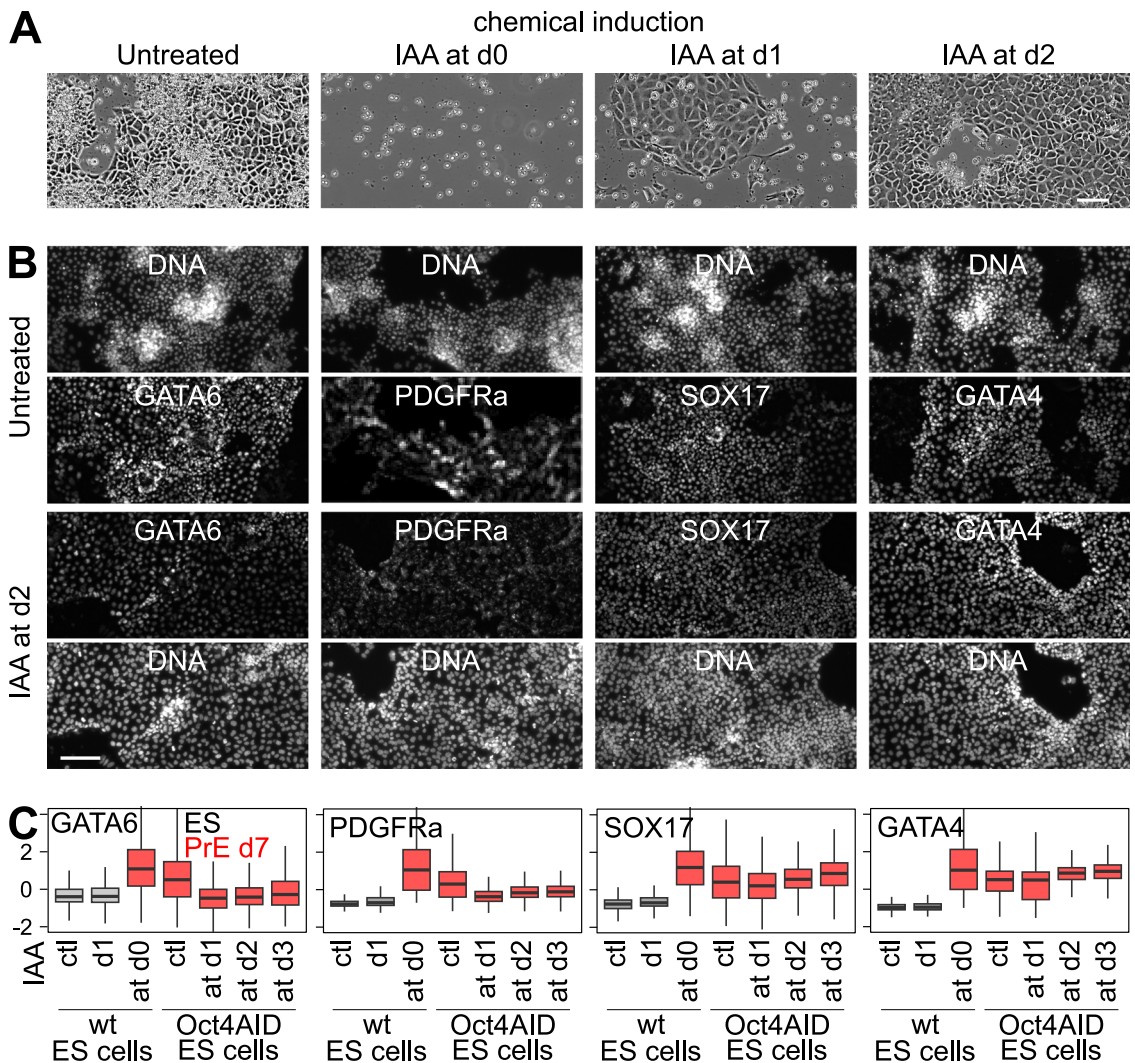

**Fig. 6 | OCT4 is essential during the early stages of PrE differentiation.**
**A** Photomicrographs of cells subject to chemical PrE induction for 7 days, either untreated or IAA-treated cells to deplete OCT4 from the onset of the chemical PrE induction (d0), from day 1 (d1) or day 2 (d2). The PrE-like cells shown for d1-treated cells represent one of the rare colonies that did not die. The experiment was repeated 2 independent times. The scale bar represents 100 μm. **B** Immunostaining of GATA6, PDGFRa, SOX17 and GATA4, after 7 days of chemical PrE induction, for untreated (top) and IAA-treated cells from day 2 onwards (bottom). The scale bar represents 150 μm. **C** Quantification of all immuno-stainings in control ES cells (ctl in gray), after 1 day of IAA treatment (d1 in gray), and in cells differentiated for

7 days (in red), either wild-type to control for IAA effects or in Oct4-AID cells in the absence of IAA (ctl) or having experienced OCT4 depletion from day 1, day 2 or day 3 onwards. The boxplots show Z-score fluorescence measurements (median−bar; 25-75% percentiles−box; 1.5-folds the inter-quartile range−whiskers; the number of cells analyzed for each factor and condition are, in the same order as in the box-plots, for GATA6: 96243, 78118, 53408, 78178, 14895, 44722, 56419; for PDGFRa: 33965, 38644, 29457, 38204, 25878, 36252, 45315; for SOX17: 78506, 79183, 58912, 64119, 25878, 42948, 62884; for GATA4: 53214, 41156, 31082, 44788, 22362, 41880, 41470, all derived from 2 independent experiments). Source data are provided as a Source Data file.

binding is faster and more robust, and its effects on nucleosomes and accessibility greater, when occurring in cooperation with other TFs. Even though GATA6's effects on chromatin manifest as soon as it binds, one to two days are needed to fully fragilize the nucleosome it targets, position the flanking ones as an ordered array, and fully open up the chromatin. This latency could be dependent on the need for DNA replication, which disrupts nucleosomes[66] even when targeted by pioneer TFs[67], but its correlation with the recruitment of other pioneer TFs indicates that GATA6 initiates a process that is fully culminated by cooperative activities. Hence, cooperative interactions between pioneer TFs appear instrumental in the way GATA6 unfolds the gene expression changes that lead to PrE differentiation.

Among the pioneer TFs involved in GATA6 function, we show that OCT4 plays a key role, as suggested more than 20 years ago[36,39]. Even though it has been debated whether its role is cell autonomous[37] or not[38], OCT4 contribution to PrE differentiation, while clear, had

remained mechanistically enigmatic until now. Here, we show that OCT4 is both an activator of *Gata6* by stimulating *Gata6* autoactivation and a mediator of its early effects, since it is redirected by GATA6 to maximize its own and SOX17 activity and readily enact differentiation. This scenario, strictly reciprocal to the redistribution of somatic TFs upon OCT4/SOX2-induced reprogramming back to pluripotency[49], is different than previous propositions for a role of PTFs in priming, extending or restoring developmental potential during lineage commitment[22,34,46–48,68]. Moreover, our observation that in the early absence of OCT4 during PrE differentiation, cell viability is compromised, opens non-mutually exclusive interpretations of its relevance. On the one hand, it is possible that GATA6-OCT4/SOX2 activate expression of genes directly involved in cell survival−anti-apoptotic genes (*Bcl2l1/Bcl-xl*, *Birc5/Survivin* and *Birc6/Bruce*) and genes with a role in PrE survival[69,70] (*Pdgfra*, *Akt1* and *Akt2*) are upregulated upon GATA6 induction (Source Data 1), with PDGFRA expression displaying

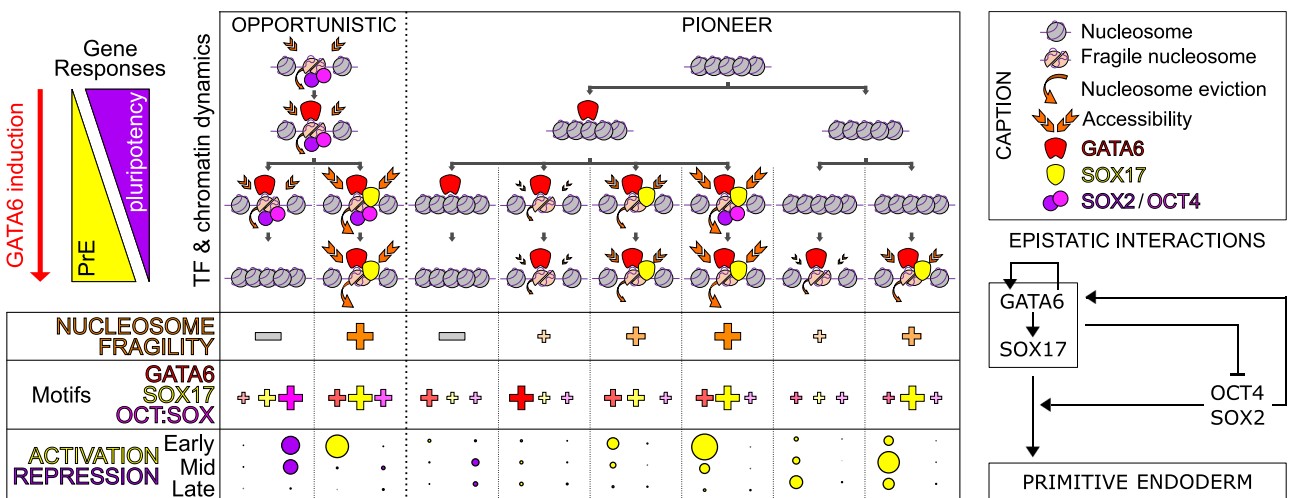

**Fig. 7 | Direct and indirect functional interactions mediated by GATA6.** Summary of the molecular observations reported in this study, highlighting different modes of GATA6 binding (opportunistic or pioneer), different dynamics (early versus late and stable versus transient) and different outcomes regarding: recruitment of SOX17 or of OCT4/SOX2; chromatin structure; gene regulatory influences at early, mid or late responsive genes; the occurrence of DNA motifs. A caption and a diagram of the epistatic interactions emerging from kinetic analyses and OCT4-depleted cells are shown.

an attenuated increase in OCT4-depleted cells. On the other hand, the early loss of OCT4 may lead to a molecular context of inadaptation between the signals pushing towards the PrE, the propensity of *Oct4-/-* cells to differentiate along the trophectoderm[36,39], and their failure to activate *Gata6*, resulting in cell death.

In summary, we show here that GATA6 takes over a multitude of gene regulatory processes, taking advantage of both the *cis*-landscape of DNA binding motifs and pre-established chromatin states, as well as the *trans*-compendium of available pioneer TFs. This enables GATA6 to multitask and to organize its activity over time, such that it can repress pluripotency, timely initiate pioneering activities at new regulatory elements required for successive waves of PrE gene activation and promote new TFs to higher hierarchies to extend its effects beyond its direct binding sites. Thus, even though pioneer TFs are especially attractive for their unique capacity in licensing close regulatory elements[71], their importance in developmental regulation goes beyond this key structural property. They fully rewire gene regulatory networks to endow the transcriptional changes required to acquire new cell identities.

## Methods
### Cell culture
**General cell culture conditions.** *Gata6*-inducible ES cells were maintained in ES medium: DMEM (Life Tech 31966047) supplemented with 15% FBS (Life Tech 10270106), 10 ng/mL LIF (Miltenyi Biotech 130-099-895) and 100 µM 2-mercaptoethanol (Gibco #31350-010)) on mitomycin C-treated (Sigma M4287) primary mouse embryonic fibroblasts (feeders). OCT4-AID iPS cells[60] were grown in serum-free 2i-containing medium: 1 µM PD0325901 and 3 µM CHIR99021 (Axon1408 & Axon1386 respectively), 0.5× DMEM/F12 (Gibco, Cat# 31331093), 0.5× Neurobasal (Gibco, Cat# 21103049), 1× N2 supplement 100× (Gibco, Cat# 17502048), 1× B-27 supplement 50× (Gibco, Cat# 17504044), 10 µg/ml Insulin (Sigma, Cat# I1882-100MG), 2 mM L-Glutamine (Invitrogen, Cat# 91139), 37.5 µg/ml BSA (Sigma, Cat# A3311-10G), 100 µM 2-mercaptoethanol (Gibco, Cat# 31350-010), 10 ng/ml recombinant LIF (Miltenyi Biotech, Cat# 130-099-895). XEN cells were cultured in DMEM-15% FBS on feeders.

**Derivation of Gata6-inducible ES cells.** $2 \times 10^6$ KH2 ES cells[72] were lipofected with 5 µg of PBS31 vector containing the GATA6-3×HA ORF (Ensembl *Gata6-201*, ordered from Genescript) and 2.5 µg of pCAGGS-FLPe (Gene Bridges). The day after, cells were trypsinized, and $3.5 \times 10^6$

were seeded on hygromycin-resistant feeders with 200 µg/mL hygromycin B. Selection was maintained for 10 days, individual antibiotic-resistant clones were picked, genotyped by PCR and tested for GATA6 induction using 1 µg/mL Dox. Two independent clones were selected for further analyses (G6.1 and G6.3). Both clones are available upon request.

***Oct4* knock-down.** 5 million of Gata6-inducible ES cells (#G6.3) were nucleofected with 200 pmol of siRNA using the Amaxa nucleofector II and Mouse Embryonic Stem Cell Nucleofector Kit (Lonza #VPH-1001—program A30) and immediately plated in ES cell media containing 1 µg/mL Doxycycline on MEFs. *Oct4* targeting siRNAs (ON-TARGETplus Mouse Pou5f1 Smartpool siRNA, Dharmacon L-046256-00-0005) were compared to untargeted control siRNAs (ON-TARGETplus Non-targeting Control Pool siRNA, Dharmacon D-001810-10-05). Knock-down efficiency was verified by qPCR, and cells were harvested for analysis 48 or 96 hours after nucleofection.

**PrE differentiation.** *Gata6*-inducible cells were seeded at 100,000 cells/cm² on feeders and differentiation was induced by the addition of 1 µg/mL doxycycline in ES cell medium, which was changed daily. Chemical differentiation was induced as previously described[58,59]. Cells were seeded at 37,000 cells/cm² (ΔEa E14Tg2A and EKOiE cells) or 74,000 cells/cm² (OCT4-AID iPS cells) onto poly-L-ornithine/laminin-coated wells of Ibidi µ-slides (#80446) in endoderm medium: RPMI 1640 Medium, GlutaMAX™ Supplement (Gibco #61870036), 2% B-27 minus insulin (Gibco #15285074) and 100 µM 2-mercaptoethanol (Gibco #31350-010). The day after, the medium was supplemented with 100 ng/ml activin A (R&D # 338-AC-010), 3 µM CHIR99021 (Axon #1386) and 10 ng/ml LIF. The medium was changed every day.

### Imaging
**Bright-field microscopy.** Cell culture pictures were taken on a Nikon Eclipse Ti-S inverted microscope equipped with: CFI S Plan Fluor ELWD ×20 objective; 89 North PhotoFluor LM-75; Hamamatsu ORCA-Flash 4.0LT camera; NIS Elements 4.3 software. Representative images of more than two independent cultures are shown.

Immunofluorescence of Gata6-inducible ES cells: Cells grown on Ibidi µ-slides (#80446) were fixed in PBS-4% PFA (Electron Microscopy Science #15700) for 20 min. Fixed cells were washed three times in PBS 1×, permeabilized in PBS-0.1% Triton-X-100 (Sigma #T8787) for 10 min and blocked for 20 min in PBST−5% BSA (Sigma #A331). Primary

antibody incubation was performed overnight at 4 °C with the following antibodies diluted at 1:1000: goat polyclonal anti-GATA6 (R&D Systems AF1700), mouse anti-OCT4 (clone C10, Santa Cruz #5279), rabbit polyclonal anti-SOX2 (Active Motif #39823), mouse monoclonal anti-ESRRB (Perseus Proteomics, H6-705-00), goat polyclonal α-SOX17 (R&D Systems AF1924–Fig. S1) or Rabbit Monoclonal α-SOX17 (Abcam #ab224637) and rat monoclonal anti-PDGFRA (APA5, Thermo Fisher #14-1401-82). Secondary antibody incubation was carried out at 1:1000 dilution for 2 hours at RT, with appropriate Alexa Fluor®-conjugated Donkey IgG antibodies (Jackson Immunoresearch). Cells were imaged on a Zeiss LSM800 confocal microscope. Representative images of more than two independent differentiations are shown.

**Immunofluorescence upon PrE chemical conversion.** Cells differentiated for 7 days were processed into their respective well, washed with PBS 1×, fixed with freshly prepared PFA 4% for 10 min at room temperature, washed 3 times in PBS 1×, and used for immunostaining. Immunostaining was performed after permeabilization with cold PBS 1×/0.5 % v/v Triton X-100 for 5 min, two washes with cold PBS 1×, and 30 min incubation on ice with PBS 1×/1% Donkey Serum (Sigma, Cat# D9663). Cells were incubated overnight at 4 °C with primary antibodies diluted 1:200 in PBS 1×/1% Donkey Serum: goat polyclonal anti-GATA6 (R&D Systems AF1700), rat monoclonal anti-PDGFRA (APA5, Thermo Fisher #14-1401-82), Rabbit Monoclonal anti-SOX17 (Abcam #ab224637), goat polyclonal anti-OCT4 (R&D systems #AF1759), goat polyclonal anti GATA4 (Santa Cruz #sc1237), goat polyclonal anti-OCT4 (Santa Cruz N19 #sc8628), rat monoclonal anti-NANOG (eBioscience #eBIOMLC-51) or rabbit polyclonal anti-NANOG (Cell Signaling Technology #CS8822S). After three washes in cold PBS 1×, cells were incubated with secondary antibodies diluted in PBS 1×/1% Donkey Serum, washed three times in PBS 1× and nuclei counterstained with DAPI (Vectorlabs, Cat#H-1200). Imaging was performed with an inverted Nikon Eclipse X microscope equipped with: X20/0.45 (WD 8.2-6.9) objective; LUMENCOR excitation diodes; Hamamatsu ORCA-Flash 4.0LT camera; NIS Elements 4.3 software. Cell Profiler[73] was used for quantifications.

### Gene expression analyses
**RNA-seq.** *Gata6*-inducible ES cells were trypsinised at each indicated time point and feeders were adsorbed on non-gelatinized plastic dishes for 20 min. Cells were washed once in PBS 1× and lysed in TRIzol. Total RNAs were extracted using chloroform, repurified by acidic phenol-chloroform extraction and precipitated. Stranded, directional polyA selected RNA-seq libraries were constructed and sequenced (paired-end 150 bp reads) on a NovaSeq6000 system (Illumina) by Novogene. Three independent replicates were used per condition except for XEN cells, for which two were used.

**Alignments and quantification.** Stranded paired-end reads were aligned to the mouse genome (mm10) using STAR[74], quantified by RSEM[75] with options " --calc-pme --calc-ci --estimate-rspd --forward-prob 0.0 --paired-end", and counts converted into Transcripts Per Million (TPM, Source Data 1) after omitting outliers identified by Principal Component Analysis (>30 standard deviations from the mean for PC1).

Differentially expressed genes (DEGs): RSEM estimated read counts per sample were rounded and used in DESeq2[76] without independent filtering. For Gata6 induction assays, all genes with FDR < 0.05 and absolute log2 fold-change >1 in at least one time-point compared to untreated cells (d0) were considered as DEGs. DEGs complying with these filters at 16 h or 24 h onwards were considered Early_UP/DOWN; at d2 or d3 onwards, Mid_UP/DOWN; at d4 and d7 or only at d7, Late_UP/DOWN. The remaining DEGs displaying more complex behaviors regarding FDR/FC filtering (e.g., FDR < 0.05 at non-consecutive time-points) were clustered based on their expression trends identified using

K-means (R function k means() with options $k = 6$, nstart = 50, iter.-max = 50). The average profile of each cluster was visualized, and the genes were attributed to the closest Early/Mid/Late_UP/DOWN group. For *Oct4* knockdown assays, all genes with FDR < 0.05 and absolute Fold-Change >0.5 were considered as DEGs; they were further filtered to belong to the list of *Gata6*-responsive genes.

**Developmental annotation.** Correlations to developmental gene expression were made using a published study of mouse embryos around gastrulation[52], with the processed data downloaded from ftp. ebi.ac.uk/pub/databases/scnmt_gastrulation.

**Correlations to TF binding sites.** Enrichments of each group of DEGs in proximity to different categories of TF binding sites were calculated using right tail *p* values of Fisher's Exact tests for the genes of each DEG cluster present within x bp of each TF binding site under consideration, for x in [1:1e + 9] bp, compared to background associations obtained from all considered genes.

### Chromatin assays
Chromatin Immunoprecipitation and sequencing (ChIP-seq): Cells were trypsinized, feeder adsorbed, and crosslinked at ~5 × 10⁶ cells/mL in PBS 1× supplemented with 2 mM DSG (Sigma-Aldrich, 80424) for 50 min followed by 10 min in PBS 1× 1%PFA (Electron Microscopy Science #15700). Crosslinking was stopped with 0.125 mM glycine, and cells were pelleted and washed once with cold PBS 1×. Crosslinked cells were either used directly for chromatin preparation or frozen in liquid nitrogen and stored at −80 °C for later processing. Fixed cells were resuspended in 2 ml of swelling buffer (25 mM Hepes pH 7.95, 10 mM KCl, 10 mM EDTA, 0.5% IGEPAL and 1× Roche protease cocktail inhibitor) and incubated on ice for 20 min. The suspension was passed >30 times in a douncer. Nuclei were centrifuged (250 × g, 15 min at 4 °C) washed and resuspended in TSE150 (0.1% SDS, 1% Triton-X100, 2 mM EDTA, 20 mM Tris-HCl pH8, 150 mM NaCl, 1× PIC) at ~3 × 10⁴ cells/mL. Samples were sonicated in 1.5 ml tubes (Diagenode) using a Bioruptor Pico (Diagenode) for eight cycles (30 s ON/30 s OFF). After centrifuging at max speed for 15 min, an aliquot was used to check the sonication efficiency and the chromatin was stored at −80 °C for further use. Immunoprecipitation was performed with 12.5 μg of chromatin per immunoprecipitation, after pre-clearing in TSE150 containing 50 μl of protein G Sepharose beads (Sigma-Aldrich #P3296) 50% slurry blocked with 0.5 mg/mL BSA and 1 mg/mL yeast tRNA (Roche #10109495001) on a rotating wheel for 3 hours at 4 °C. Pre-cleared chromatin was incubated at 4 °C overnight rotating on-wheel with the following antibodies: anti-GATA6 (Cell Signaling technology #5851); anti-OCT4 (abcam ab19857); anti-SOX2 (Active Motif 39823); anti-ESRRB (Perseus Proteomics, H6-705-00); anti-SOX17 (R&D Systems AF1924). A 20 μL aliquot was reserved for input, and 50 μL of blocked protein G beads were added for 1 h 30 mins at 4 °C on-wheel. Beads were pelleted and washed on-wheel with 1 mL cold buffer in the following order: three washes in TSE150, 1 wash in TSE500 (TSE150 but with 500 mM NaCl), 1 wash in washing buffer (10 mM Tris-HCl pH8, 0.25 M LiCl, 0.5% NP-40, 0.5% Na-deoxycholate, 1 mM EDTA) and 2 washes in TE. ChIP was eluted in 100 mL elution buffer (1% SDS, 10 mM EDTA, 50 mM Tris-HCl pH 8) and beads rinsed in 150 mL TE-1% SDS. The eluate and supernatant were pooled. Input and precipitated chromatin were reversed-crosslinked overnight (65 °C in TE−1% SDS with 100 μg stabilized Proteinase K; Eurobio #GEXPRK01), phenol-chloroform-extracted and ethanol-precipitated. Libraries were constructed using the NEB-Next Ultra II DNA library prep kit (NEB #E7645) using 0.6 μM annealed adapters for ligation. Custom designed Y/forked adapters based on Illumina Truseq indexes with 6-8nt unique molecular identifiers (UMI) 3' of the index sequence were used[57]. Libraries were amplified by 8 to 12 PCR cycles, purified using SPRI beads, quantified with a Qubit 3 (Invitrogen) and fragment size distribution checked on an Agilent 2200

Tapestation. Pooled libraries were sequenced (single-end 75 bp reads) on a Nextseq2000 sequencing system (Illumina), except *Oct4* knock-down libraries, which were sequenced in PE (100). For most conditions, two independent chromatin preparations were analyzed, except for GATA6 d0 ($n = 1$), OCT4 XEN ($n = 1$).

**Assay for transposase-accessible chromatin using sequencing (ATAC-seq).** 100,000 cells were washed in PBS 1×, pelleted and resuspended on ice in 50 μL transposition reaction mix (25 μL 2× TD Buffer, 2.5 μL Tn5 Transposase, 22.5 μL nuclease-free H2O, Illumina Nextera DNA library prep kit). The transposition reaction was incubated at 37 °C for exactly 50 min and stopped by the addition of 250 mL of binding buffer (Qiagen MinElute cleanup kit). Transposed DNA was purified with the MinElute cleanup kit (Qiagen #28204) and resuspended in 10 μL of $H_2O$. Libraries were prepared by adding 2.5 μL 25 μM primer Ad1.noMX, 2.5 μl 25 μM Ad2 indexing primer[77], 9 μl Nuclease-Free Water, 1 μl of a 1:8 dilution of Quant-iT Picogreen dye (Invitrogen, # P11496) and 25 μl of KAPA HiFi HotStart 2× Master Mix (Kapa Bioscience #KK2502) to the transposed DNA. Amplification was carried out in a LightCycler II qPCR machine (Roche), and fluorescence was monitored to stop amplification during the exponential phase. Libraries were purified using SPRI beads, quantified using a Qubit 3 (Invitrogen) and fragment size distribution checked on an Agilent 2200 Tapestation. Pooled libraries were sequenced (PE150 reads) on a NovaSeq6000 (Illumina) by Novogene. All ATAC-seq were prepared from three independent cell cultures except for d4 ($n = 2$).

**Micrococcal nuclease sequencing (MNase-seq).** Cells were fixed in warm ES medium with 1% PFA (Electron Microscopy Science #15700) for 10 min at room temperature and quenched with 0.125 M glycine. Cells were pelleted and washed in ice-cold PBS 1×. $1.25 \times 10^6$ fixed cells were resuspended in 500 μL of MNase buffer (50 mM Tris-HCl pH8, 1 mM CaCl2, 0.2% Triton X-100, 1× Roche Protease cocktail inhibitor) and prewarmed for 10 min at 37 °C. 16 U of MNase (NEB Cat# M0247) was added to the reaction and digestion was carried out for exactly 10 min at 37 °C with low mixing on a ThermoMixer (Eppendorf #5384000020). The reaction was stopped on ice by the addition of 500 μl of 2× STOP buffer (2% Triton X-100, 0.2% SDS, 300 mM NaCl, 10 mM EDTA). Tubes were placed overnight on a rotating wheel at 4 °C to allow diffusion of the digested fragments. The suspension was spun down, and the supernatant frozen at −80 °C. 25 μL of digested chromatin was reversed-crosslinked overnight (see ChIP-seq), phenol-chloroform-extracted and ethanol-precipitated. Samples were resuspended in TE, incubated with 2 μL of RNAse (Roche #11119915001) for 30 min at 37 °C and purified with SPRI beads. DNA concentration was measured using a Qubit 3 (Invitrogen) and size-checked on an Agilent 2200 Tapestation. Libraries were prepared from 20 ng purified DNA: samples were end-repaired, A-tailed for and ligated to 1.25 μL of 0.2 μM annealed custom adapters as previously described[57]. Libraries were amplified by qPCR, and amplification stopped during the exponential phase. Library concentration was measured using a Qubit 3 (Invitrogen), and fragment distribution was checked on an Agilent 2200 Tapestation. Pooled libraries were sequenced (PE150 reads) by Novogene on a NovaSeq6000 (Illumina). All MNase-seq assays were performed on three independent cell cultures.

**Bioinfomatic analyses of chromatin-based datasets**
**Reads processing.** Libraries were demultiplexed and fastq files generated (including for index reads for ChIP and MNase-seq libraries) using bcl2fastq conversion software (Illumina). Adapters were trimmed using cutadapt[78] Trimmed reads were aligned with bowtie 2[79] to the mm10 genome using the parameters --local --very-sensitive-local --soft-clipped-unmapped-tlen. Optical duplicates were removed using SAMtools[80] and bams filtered for MAPQ > 30. Reads of bowtie2-aligned SAM files were grouped by name using SAMtools collate. Mate

coordinates and insert size fields of PE reads were filled using SAMtools fixmate. UMIs were added to the SAM RX tag using AnnotateBamWithUmis (https://fulcrumgenomics.github.io/fgbio/). Duplicate reads were marked with SAMtools markdup and filtered. For ATAC-seq, reads were shifted inwards by 4 bp to plus-stranded insertions and −5 bp to minus-stranded insertions, as recommended[77].

**Peak-calling.** ChIP-seq peaks were called against inputs using MACS2[81] with a $q$ value cutoff of 0.01 except for GATA6 ($10^{-4}$) and ESRRB (0.05). For each factor and time-point, the BAMs of all replicates were merged before peak calling and the peaks were retained if they were also called in all individual replicates.

**Peak quantification.** Quantifications were performed with the bamsignals R package (https://github.com/lamortenera/bamsignals) with systematic correction to the library sizes and counting the number of reads either falling into peak coordinates (Source Data 1) or covering each base of a 4kb window centered on the peak summit, defined as the max GATA6 signal for GATA6 peaks or the max ATAC-seq signal for pluripotency TF peaks not bound by GATA6. To identify GATA6 summits, the reads were extended as estimated by MACS. For ATAC-seq, PE fragments <100 bp were used to measure accessibility and identify the base with the maximum signal. Quantifications of histone modifications to characterize GATA6 peaks were performed using ES cells' bam files downloaded from Encode.

**Oct4 knock-down.** To identify OCT4-sensitive regions, all GATA6 peaks were quantified 48 h after simultaneous GATA6 induction and *Oct4* or Ctl siRNA-mediated knockdown. DESEQ2 was used to identify regions displaying significant changes of GATA6 binding between *Oct4* and Ctl knockdown (FDR < 0.1 and absolute log2 Fold-Change > 0.5).

**DNA motifs.** Motif occurrences and statistics were obtained by scanning GATA6 peaks using FIMO[82] for the following motifs obtained from JASPAR database: MA1104.1 (GATA6), MA0142.1 (OCT4:SOX2), MA0078.1 (SOX17), MA0141.3 (ESRRB).

**Analysis of nucleosomes.** Nucleosomes were mapped either using ATAC-seq or MNase-seq datasets. For ATAC-seq, only fragments of 140:250 bp were considered; for MNase-seq, only fragments of 130:160 bp were considered. The coverage of the signal at specific regions was calculated using fragment midpoints exclusively, normalized to the library depth. The average profiles were smoothed using the loess() function in R, with a span of 0.05 for ATAC-seq and 0.07 for MNase-seq. For MNase-seq data, the profiles were internally normalized to the flanking sides (the average of the 1:500, 3501:4000 intervals was subtracted from the signal). These profiles were used to quantify nucleosome order, using spectral densities (calculated in R using the spectrum() function and a period of 180 bp, which gave maximal signal), and nucleosome densities on GATA6 summits, computed by averaging the signal in 200 bp around the summit.

**Categorical variables and statistics.** To generate the groups of TF binding regions, we used k-means clustering implemented as described before: to split the regions as Early&Constant, Transient or Late GATA6 binding sites, we used quantitative data of every time-point of GATA6 enrichment normalized to the corresponding inputs and log2 transformed. For SOX17 (bound by or not) and for pluripotency TFs (bound at d0, ectopically or not at all), we used binary peak calling data. To characterize the regions according to histone modification signatures, we used library size normalized, region width normalized, and log2 transformed counts, which were subsequently Z-scored for each individual histone mark to scale absolute levels between different marks. The relationships between variables were assessed with the mosaicplot() function in R followed by two-sided Chi-square tests and

 

Fisher Exact tests to compute the odds ratios and associated *p*-values of specific depletions/enrichments. To test the prediction of GATA6 groups based on different variables, we used the Python version of LightGBM[54] for multiclass classification (with max_depth = 10, num_leaves = 180, learning rate = 0.001, lambda_l1: 9.18e-05). Hyperparameters were tuned using a bayesian hyperparameter tuning framework optuna with 5 fold cross-validation (StratifiedKFold(n_splits=5, shuffle=True)). Number of estimators was corrected according to the early stopping option with *n* = 100 and evaluation metric eval_metric = 'multi_logloss'. ROC curve was calculated for the multiclass classification by binarizing the output and implementing macro-averaged one-vs-rest strategy. Feature importance was estimated using pre-build lightgbm attribute feature_importance_ with default parameter importance_type = 'split'.

### Reporting summary
Further information on research design is available in the Nature Portfolio Reporting Summary linked to this article.

### Data availability
The RNA-seq, ATAC-seq, MNase-seq and ChIP-seq data generated in this study have been deposited in the Gene Expression Omnibus database under accession code GSE268719. https://www.ncbi.nlm.nih.gov/geo/query/acc.cgi?acc=GSE268719 The MNase-seq data with different MNase concentrations used in this study are available in the Gene Expression Omnibus database under accession code GSE122589. https://www.ncbi.nlm.nih.gov/geo/query/acc.cgi?acc=GSE122589 Source data are provided with this paper.

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

## Acknowledgements

The authors acknowledge Jose Silva and Lawrence Bates for the kind gift of *Oct4*-AID cells, Jerome Artus for XEN cells, all members of the laboratory for critical discussions and the OMICs facility of Institut Pasteur for access to sequencing platforms. This study was funded by the Labex Revive (ANR-10-LABX-73; R.X.C., M.C.T., P.N.), the CNRS (M.C.T. and P.N.), the Institut Pasteur (P.N.), the ANR (ANR-14CE11-0017PrE-piSpec; M.C.T.) and the European Research Council (ERC-CoG-2017 BIND; P.N.).

## Author contributions

This study was conceived by P.N. and M.C.T. Experiments were designed and executed by R.X.C. and A.D. (chemical PrE induction), with help from I.G. and S.V.P. Gene expression analyses were performed by A.C. and P.N.; chromatin analyses were performed by R.X.C. and P.N. The paper was written by R.X.C., M.C.T. and P.N. and approved by all authors.

## Competing interests

The authors declare no competing interests.
