## [Transparent Peer Review file · Nature Communications]

Molecular and epistatic interactions between pioneer transcription factors shape nucleosome dynamics and cell differentiation

Corresponding Author: Dr Pablo Navarro

Version 1:

Reviewer comments:

Reviewer #1

(Remarks to the Author)

The revised manuscript satisfactorily addresses my previous concerns. The more comprehensive Oct4 loss-of-function analyses are particularly valuable, as they provide compelling and novel mechanistic insights into its role in modulating PrE pioneer factor activity and PrE gene expression.

Reviewer #2

(Remarks to the Author)

Coux et al. have extensively revised their manuscript to provide a more straightforward story that now includes some experimental investigations of the GATA6-induced process of primitive endoderm differentiation. Indeed, they provide evidence that knocking down OCT4 expression alters the process of PrE differentiation through GATA6 and SOX17 targets. Altogether, the paper now provides a coherent story where the extensive data are integrated into a mechanistic outlook.

I do not find good justification for the use of the word "pioneerome" because there is no evidence for the formation of a corresponding protein complex. In that respect, the wording on L.389 of "formation of a pioneerome" is inappropriate as the authors have not documented any sort of complex. Since cooperative recruitment of different transcription factors at enhancers is an old concept, there is no justification in this reviewer's opinion to introduce a new term in the context of cooperative recruitment of pioneer factor at enhancers.

Reviewer #3

(Remarks to the Author)

The revised version of the manuscript by Coux and colleagues has significantly improved due to the extensive changes and adaptations the authors made.

By excluding their data regarding the role of nuclear receptors in the PrE differentiation, the authors have made their manuscript more coherent and streamlined. They focus now solely lies on the role of TF-TF interplay at various stages of PrE differentiation.

Adaptations to main figures such as moving certain subfigures to the supplement have helped to better follow the text and logic of the manuscript. Adding experimental dissection of the role of OCT4 in PrE differentiation underlines the interplay between GATA6 and cell type-specific TFs and helps elevate the paper past the abundant but solely associative evidence shown in the original manuscript.

Following the various subclassification of GATA6, OCT4, and SOX17 binding sites at different stages of the differentiation process remains complicated, but making these distinctions and understanding them is the entire point of the paper. The main conclusion I draw from this study is that a pioneer TF that reprograms cell identity builds on the set of key transcription

factors present in the cell and transform their binding over time. The OCT4-KD experiments also suggest that this process is dependent on the key TFs present in the cell type that is modified by the pioneer TF. As such, I will cite this study for an in-depth description of this process, that I think has not been described in this detail before. I support the publication of the revised version. However, I will not adapt the term “pioneerome”, but that is my problem.

Point by point response

We thank our reviewers for acknowledging the efforts we made to address their concerns and to validate the improvements made.

Reviewer #1 (Remarks to the Author):

The revised manuscript satisfactorily addresses my previous concerns. The more comprehensive Oct4 loss-of-function analyses are particularly valuable, as they provide compelling and novel mechanistic insights into its role in modulating PrE pioneer factor activity and PrE gene expression.

We thank Reviewer 1 for the positive evaluation of our improvements and changes

Reviewer #2 (Remarks to the Author):

Coux et al. have extensively revised their manuscript to provide a more straightforward story that now includes some experimental investigations of the GATA6-induced process of primitive endoderm differentiation. Indeed, they provide evidence that knocking down OCT4 expression alters the process of PrE differentiation through GATA6 and SOX17 targets. Altogether, the paper now provides a coherent story where the extensive data are integrated into a mechanistic outlook.

I do not find good justification for the use of the word “pioneerome” because there is no evidence for the formation of a corresponding protein complex. In that respect, the wording on L.389 of “formation of a pioneerome” is inappropriate as the authors have not documented any sort of complex. Since cooperative recruitment of different transcription factors at enhancers is an old concept, there is no justification in this reviewer’s opinion to introduce a new term in the context of cooperative recruitment of pioneer factor at enhancers.

We thank Reviewer 2 for the positive evaluation of our improvements and changes. We have removed the term pioneerome from the paper, as suggested.

Reviewer #3 (Remarks to the Author):

The revised version of the manuscript by Coux and colleagues has significantly improved due to the extensive changes and adaptations the authors made.

By excluding their data regarding the role of nuclear receptors in the PrE differentiation, the authors have made their manuscript more coherent and streamlined. Their focus now solely lies on the role of TF-TF interplay at various stages of PrE differentiation.

Adaptations to main figures such as moving certain subfigures to the supplement have helped to better follow the text and logic of the manuscript. Adding experimental dissection of the role of OCT4 in PrE differentiation underlines the interplay between GATA6 and cell type-specific TFs and helps elevate the paper past the abundant but solely associative evidence shown in the original manuscript.

Following the various subclassification of GATA6, OCT4, and SOX17 binding sites at different stages of the differentiation process remains complicated, but making these distinctions and understanding them is the entire point of the paper. The main conclusion I draw from this study is that a pioneer TF that reprograms cell identity builds on the set of key transcription factors present in the cell and transform their binding over time. The OCT4-KD experiments also suggest that this process is dependent on the key TFs present in the cell type that is modified by the pioneer TF. As such, I will cite this study for an in-depth description of this process, that I think has not been described in this detail before. I support the publication of the revised version. However, I will not adapt the term “pioneerome”, but that is my problem.

We thank Reviewer 3 for the positive evaluation of our improvements and changes. We have removed the term pioneerome from the paper, as suggested.